# Soil Moisture a Posteriori Measurements Enhancement Using Ensemble Learning

**DOI:** 10.3390/s22124591

**Published:** 2022-06-17

**Authors:** Bogdan Ruszczak, Dominika Boguszewska-Mańkowska

**Affiliations:** 1Department of Computer Science, Opole University of Technology, 45-758 Opole, Poland; 2Potato Agronomy Department, Plant Breeding and Acclimatization Institute—National Research Institute, 05-870 Radzików, Poland; d.boguszewska-mankowska@ihar.edu.pl

**Keywords:** soil moisture, moisture sensors, ensemble learning, machine learning, potato watering, sensor calibration enhancement

## Abstract

This work aimed to assess the recalibration and accurate characterization of commonly used smart soil-moisture sensors using computational methods. The paper describes an ensemble learning algorithm that boosts the performance of potato root moisture estimation and increases the simple moisture sensors’ performance. It was prepared using several month-long everyday actual outdoor data and validated on the separated part of that dataset. To obtain conclusive results, two different potato varieties were grown on 24 separate plots on two distinct soil profiles and, besides natural precipitation, several different watering strategies were applied, and the experiment was monitored during the whole season. The acquisitions on every plot were performed using simple moisture sensors and were supplemented with reference manual gravimetric measurements and meteorological data. Next, a group of machine learning algorithms was tested to extract the information from this measurements dataset. The study showed the possibility of decreasing the median moisture estimation error from 2.035% for the baseline model to 0.808%, which was achieved using the Extra Trees algorithm.

## 1. Introduction

In agriculture, it is essential for crop appropriate growth and quality to apply water in an optimal amount and at the right time. It is crucial in the field of irrigation and also for our daily life due to its use by industries and other environmental activities. Agriculture absorbs large amounts of water. This is why there is a need to control these amounts. Thus, water availability for irrigation purposes is limited, primarily due to the increasing demand of the world population, which is estimated to rise to about 9.8 billion in 2050. Problems related to a lack of water will likely increase if long-term global climate change predictions are correct. It has been noted that global mean land and ocean surface temperatures increased by 0.8 °C between 1888 and 2012, whereas the worldwide average surface temperature has been predicted to increase by by 1.4 to 5.8 °C by 2100. Furthermore, increases in evaporation and reductions in precipitation rates are also foreseen. It will cause a decline in water resources in the 21st century. This is why efficient irrigation systems are strongly demanded, and that kind of solution can help to decrease agricultural water consumption, improve farm profitability and reduce environmental impacts [1,2]. In recent years, advances in electromagnetic sensor technologies have made automated irrigation scheduling a reality by using state-of-the-art soil moisture sensing devices. However, correct sensor positioning and interpretation of the measurements are critical to successfully implementing these management systems [3].

Soil moisture is one of the most important parameters governing meteorological, hydrological, agricultural, and climate-related events. Thus, it is essential to have information regarding soil moisture and its spatial and temporal variations.

Soil sensors are essential for accurate soil moisture measurements necessary for successful cultivation planning. Many popular sensor implementations utilize simpler sensors (i.e., 10HS), which provide limited accuracy without individual, labor-intensive calibration of each device [4,5,6]. Alternatively, more advanced probes with some autocalibration capability and higher measurement precision can be used. Still, such devices are also several times more expensive, and their implementation for larger farms is economically inefficient and rather unprecedented in practice.

This work compared an indirect soil moisture content estimation method with the gravimetric method [7,8] that is commented using our results in the last section of this paper. This was aimed to assess the recalibration and accurate characterization of commonly used smart soil-moisture sensors using computational methods. The following study introduces a method to increase the accuracy of soil moisture measurement at the root level using simple probes without significantly increasing the cost of this method. For this issue, it was proposed to perform measurements with inexpensive measuring probes, along with the accompanying limited number of manual gravimetric reference measurements, the accuracy of which is much higher. Next, the preparation of machine learning models, which, based on a set of reference gravimetric measurements, collected measurements from many measurement probes located in different places in the field and without additional calibration, recorded data on cultivation and current meteorological data, will allow for estimating the measurements of individual probes with a lot less error compared to manual gravimetric measurements.

## 2. Materials and Methods

### 2.1. Soil Moisture Measurement

Soil water content is vital to estimate in many biophysical processes, and it can be measured directly or indirectly [9,10]. In the direct method, water content is estimated in the soil using either gravimetric or volumetric methods based on the primary measurement of mass or volume of water and soil, respectively. The volumetric method directly measures soil moisture as volumetric water content (VWC) in percentage [11]. In the gravimetric method, the sampled soil is initially dried in the oven at a temperature of 105–110 °C for 24 h and weighed to estimate the mass of dry soil. Then, the soil is wet with water, and the weight of wet soil is taken to find the amount of water in the soil to estimate gravimetric water content (GWC), which indicates soil moisture in percentage. GWC is calculated as the mass of water per unit mass of dry soil [12]. The gravimetric method is very precise [13], but it is usually destructive because the soil sample is removed from the field to be analyzed. Moreover, it is a time-consuming, laborious and impractical way of measuring soil water content in the open field. Because of these limitations, a variety of indirect measurements have been developed.

Indirect methods to provide the output use soil moisture sensors in which sensor electrical parameter changes in response to soil moisture. These commonly applied soil moisture sensors utilize indirect sensing tensiometers, electrical resistance blocks, capacitive sensors, and time-domain frequency-domain reflectometer sensors. Such sensors are widely available that provide measurements with the desired frequency, have low maintenance needs and costs, and are easily set up for an automated work. Measurements from sensors are collected and read quickly. Moreover, extensive dataset preparation becomes more accessible and effortless due to sensor usage. The proper use of such devices requires a good understanding of several factors affecting the measurement, such as the geometric properties of the sensors, soil temperature, bulk soil electrical conductivity, and the electronic features of the different sensors.

### 2.2. Soil Moisture Measured Using Sensors

During the growing season, 24 sensors 10HS (METER Company, Pullman, WA, USA) [6] were placed on 5.6 m${}^{2}$ of experimental microplots, on which potatoes were grown. Microplots had a soil profile separated by a layer of concrete to a depth of ten meters to avoid water leakage. The experiment included two soil profiles: loamy coarse sand—on 12 plots and loamy fine sand—on 12 plots. Soil humidity (VWC%) was monitored throughout the growing season using 10HS (METER Company) with a ProCHECK handheld reader.

In order to obtain differentiated soil moisture, the following combinations were used: natural weather conditions prevailing in the growing season, provocative drought achieved by placing tents on microfield sites, natural rainfall conditions supplemented with drip irrigation for about 50% of the needs, natural rainfall conditions supplemented with drip irrigation about 75% of the needs, and natural rainfall conditions supplemented with irrigation dripping approximately 100%. Each plot was a separate experimental object. Table 1 shows the application of water and soil moisture (VWC %) to each microplot. To provide various soil moisture readouts during the experiment, different watering has been applied. In Table 1, one can find the exact amount that was delivered to each microplot. It is crucial to prepare a set of measurements of high variance that will later allow proper method evaluation and preparation of a performant algorithm.

The experiment took place in the season 2020 in Jadwisin, in the facility of Plant Breeding and Acclimatization Institute—National Research Institute, Division Jadwisin. The geographical coordinates of the plots that were used in the study are: $52^{{}^{\circ}}29.00562$ N, $21^{{}^{\circ}}2.67360$ E.

### 2.3. Soil Moisture Measured Using a Gravimetric Method

The soil moisture was also done using the gravimetric method. The soil was collected, put into a container, weighed in the sampled (moist) condition, oven-dried, and weighed again after drying. Drying was done at 105 °C to constant weight. It was expressed in percent.

The soil moisture measurements were taken several times each week and for each microplot during the vegetation period from May to September, with the parallel use of the gravimetric method and the 10HS sensor method. During that period, 766 reference gravimetric measurements were collected and assigned to the sensor readouts. The sensors that were used have a volume of influence of 240 mL, and thus for the referenced hand-made sampling, a similar soil volume was collected each time.

The characteristic of the distribution of the collected measurements was examined among different categories to check if they were similar. For both measuring methods, the measurements fall into the normal distribution. The characteristics of obtained results divided according to the soil type or potato variety do not differ significantly as well. The comparison of those distributions using boxplots for categories and measurement methods is depicted in Figure 1.

It can be noticed that, although both methods of measurement returned reproducible results, unfortunately, there are significant differences between both sets of data. Measurements determined using sensors are noted on a different scale, and they require conversion before comparing them with the results from the gravimetric method. This can be done by each sensor’s individual calibration before actual usage to determine their individual conversion curve, or alternatively, as recommended by the sensor manufacturer, application of a linear conversion, using particularly the coefficient, a divisor advised by the sensor manufacturer for different types of soil. The results of such linear estimation for the collected measurements are presented in Figure 2 (additionally, in Section 3.1, the improvement of the measurement using the default measurement divider optimization is commented on). The manufacturer declares an approximate 3–4% accuracy for the devices used. The accuracy obtained for the tested set, verified with reference gravimetric measurements, was higher, and the error in the estimation of moisture in the roots was 2.545% (average estimation error for estimated moisture in relation to moisture measured using the gravimetric method). The results of the estimation of measurements using the correction method suggested by the manufacturer (named as baseline linear model) are also presented to compare the results obtained later in the article in Table 2.

The next stage of the study was an attempt to increase this accuracy. Collected reference gravimetric measurements and data on cultivation, the location of individual sensors, and recorded meteorological data near the probes could potentially improve the estimation of accurate moisture data.

Thus, to provide additional insights and to provide potentially useful information, during the experiment, meteorological conditions were monitored by the Campbell weather station (Campbell Scientific Inc.). The parameters that have been observed were: air humidity and air temperature measured at 200 cm above ground level, the ground temperature measured at depths of 5, 10, 20, and 50 cm below ground level, as well as precipitation in millimeters. All those parameters were recorded a dozen meters from the experimental plots.

In addition to the weather, other data on the experiment were recorded. They comprise the performed watering details (these data were aggregated for the next 1, 2, 3, 6, and 24 h preceding the measurement), information on the type of soil, and whether the plot was covered from rainfall. The precipitation and watering data were aggregated similarly over time.

### 2.4. Modeling Using Ensemble Machine Learning

The algorithm for the presented methodology has to allow simultaneous modeling reflecting different characteristics of each individual sensor. Therefore, for this process, several ensemble algorithms have been tested. In addition, therefore, besides measurement data, some of the features passed to the algorithm encode the individual sensors and their location. Ensemble models should work in such cases in a more performant way than simple linear models. Thus, the model should provide the precise moisture estimations for many independent, uncalibrated sensors located in various places (and, i.e., allow for reproducing different soil characteristics).

Among many others, some ensemble machine learning models are frequently advised for sensor measurement estimations. Several investigators report the excellent performance of Extra Trees and Random Forest algorithm when dealing with moisture measurements [14,15,16]. According to others, a good way of obtaining low error estimations is the application of Extreme Gradient Boosting [17,18]. Other algorithms, like Adaptive Boosting (AdaBoost) for water parameters’ estimations [19] or Light Gradient Boosting Machines for evapotranspiration modeling [20], are also recommended. For a broader explanation of the applied algorithms, one could have a look into [21] for the Extra Trees method or into [22,23] for Random Forest or Extreme Gradient Boosting implementation descriptions.

For a reference to a more advanced method, the preparation of a simple linear model could be helpful. It enables the comparison of the results, which helps to assess the final model errors. The investigation of drying phenomena using such statistical modeling was presented in [24].

Evaluating such a method requires an adequate metric that would allow proper estimation error assessment between variants and in relation to the one given by the sensor manufacturer. There are several frequently applied metrics for such an estimations: those focusing on error description, like mean absolute error (in this paper denoted as AAE to distinguish it from median absolute error, indicated as MAE), calculated as the average difference between predictions and measurements from the test set, or those providing information on the explained variation like the coefficient of determination (usually denoted as $R^{2}$) and explained variance score. Besides mean absolute error, which is good to estimate the error in natural units, for closer error investigation, one could calculate: median absolute error, which is more robust even when dealing with outliers, mean squared error (MSE), which returns non-negative values, rapidly growing for more extensive errors, mean squared log error (MSLE) that helps to deal with errors varying on different scales, root mean square error (RMSE) that brings back the metric to the natural scale, or max error (ME) that is used when highlight extreme errors are needed. Selection of the mentioned metrics using the separator test dataset should allow for assessing the studied estimation models. It is worth recommending a proper training procedure implementing cross-validation, or a prominent optimization method, to determine the appropriate model hyper-parameters, i.e., following the procedure [25] that is commonly reported as performance improvements. To prepare models for this investigation, the standard five-fold cross-validation procedure using the training set has been used.

## 3. Results

For the several experiments that are described in this section, a dataset comprising consecutive pairs of sensors readouts and reference gravimetric manual measurements has been used. Additionally, the meteorological parameters presented in Section 2.3 have been assigned to each measurement pair.

Next, the test set has been separated from the whole dataset to enable the assessment of the prepared algorithms. The test set has been built using data from 4 of all 24 experimental plots. The plots have been chosen to be included into the test set in such a configuration that the distribution of data in relation to the soil type, potato varieties, and different watering strategies was the same as in the training part of the dataset.

For the following tested models, we calculated several metrics using the test set that was not used during the training process of presented estimation models. Those metrics were: mean absolute error, median absolute error, root mean square error, and coefficient of determination.

### 3.1. Baseline Model

First, to calculate the actual soil moisture, we followed the advice of the sensor’s manufacturer. The recommendation is to use a soil type related divider to convert the readouts. This divider for the described experiment soil type is 1.8. As is pretty obvious, such a method has some undeniable limitations. It is not trivial to categorize the soil type, where one places the sensor. Even the sensor manufacturer in the documentation points out that [26]: the poor accuracy due to necessary calibration regarding the soil type and the soil salt content, and that those parameters could vary for different locations and are not easy to measure. The soil characteristic is also rather a continuous variable than a categorical one. In addition, another considerable difficulty is the necessity of this parameter assessment for each placed sensor. The spatial variance also does not aid a linear measurement estimation, and such linear calibration has to lead to an accuracy loss.

To improve the accuracy of the device, the manufacturer suggests the calibration using the coefficients for different soil types, and proposes the universal linear transformation, giving the coefficients values for several soil types [26].

The results for the correction calculated using the advised method on the collected data are presented below. The divider was selected using the device documentation according to the plot soil type (q=1.8) [26], and the median absolute error is 2.546%. We checked the influence of the divider on the correction error and tested various levels of this parameter *q*. We found that, for such a linear correction, a different level of parameter provides better performance, and for the tested soil and sensors configuration, and (q=2.065), the error was lowered by 0.38 (MAE = 2.16201). Figure 2 depicts the linear model for sensor readout correction (right side of the chart) and the estimation error in relation to various correction dividers (on the left).

The next step of the analysis attempted to address the second limitation of the rigid correction method, which uses one divider only and does not fit every sensor lowering their performance. To tackle this issue, the proposed model takes into account soil variability and individual experimental plot characteristics. Table 2, presented below, gathers the results for the three best model configurations, compared to the baseline model, for the models prepared using the following features configurations:only sensor measurements (MAE = 1.641),sensor measurements and encoded information on plot soil type (MAE = 1.545),sensor measurements and each plot number (**MAE = 1.462**), to allow the algorithm to train an individual model for each of 24 sensors and lead to the lowest error rate in the linear correction approach.

### 3.2. Additional Ensemble Models and Parameters’ Tests

Part of the experiment aimed at increasing the soil moisture estimation performance by extending its interpretation ability by feeding the regression model with additional meteorological parameters and other metadata collected during the investigation. Ensemble learning has been applied to allow model training using such a diverse dataset. To prepare the regressors for sensor calibration, the following models have been tested: Random Forest, Extreme Gradient Boosting (XGB), Light Gradient Boosting Machines (LGBM), and Extra Trees Regressor.

The application of ensemble models allows for decreasing the median absolute error to 0.831%. As one could spot (Figure 3), among all tested models’ architectures, the lowest error level has been observed for those trained using Extra Trees Regressor. Both mean and median absolute errors for the estimated soil moisture indicated the best performing set of models for that tested configuration.

During the study, many feature configurations were tested to measure their impact on the estimation model performance. Model supplementation with collected weather parameters allowed the most significant improvement. The best-resulting configuration was comprised of air humidity and ground temperatures measured at different depths. The median absolute error for estimations generated using such a configuration was only **0.831**%. It is essential to add that the weather measurements had been collected in one meteorological station only, in contrast to the soil moisture measurement that was done separately for each of the 24 experimental plots. Thus, the algorithm was able to capture the trends in weather data over time, but this trend was not different between samples for individual sensors. All model configuration results are listed in Table 3.

The following experiment aimed to test if the information on watering and observed precipitation could influence estimation performance. In Figure 4, one could notice almost no difference in results for models trained using the watering data. The features added to the training dataset one by one were comprised of data aggregated for the last 1, 2, 3, 6, or 24 h and have been compared to the model trained on a set without these data. Such additional features do not significantly improve the moisture estimation.

The best performance has been achieved for the model architecture based on the Extra Trees, with access to the information on watering during the last hour. That model resulted in the mean absolute error of 1.162%, and even better and less prone to outlier measurements, the median absolute error of 0.828%. A detailed compilation of the tested estimation models for this series of experiments is presented in Table 4. The procedure required to achieve the eventual estimations for probe readouts, besides those readouts themselves (a model feature called: moisture probe), requires providing more data. Feeding the calculation with air temperature, air humidity, and soil temperature (model features: H200, T200, Tg5, Tg10, Tg20, Tg50) is necessary. It also needs values related to the exact experimental plot, like field number that identifies the probe, type of soil, and watering treatment type (model features: field, soil type, watering type, houses).

A piece of interesting information occurs when one looks closely at the features’ importance values. As expected, the most important is the variable denoting probe readout (moisture probe) whose value is 57.4%. Similarly important are the watering type (8.6%) and soil temperature at 50 cm depth (Tg50 = 7.7%). Several other model parameters range from 4–5% (Tg20 = 4.9%, Tg10 = 4.6%, Tg5 = 4.2%), and some others with the lower value. However, surprisingly low importance was related to the soil type mapping feature (soil_type), with the lowest value equal to only 0.8%.

Part of this experiment examined feature encoding observed rainfall and its influence on the model performance. Similar to the previous checks, we tested different time-frames for aggregating rainfall data. Models with precipitation summed up in the prior hour, 2, 3, 6, or 24 h were evaluated, and one without this feature, but their performance did not let us improve overall estimator performance. The slightly better results were noted for a model trained using Extra Trees and the 6-hour precipitation information with a median absolute error of 0.808%. For the models based on Extra Trees, results were found better for the investigated validation set. This could mean that the algorithm had a lower tendency to overfit to training data, and in this case has better generalization ability.

### 3.3. Required Calibration Procedure

To perform the last experiment, the whole collected measurements dataset of measurement pairs (sensor readouts and manual reference measurements) was consecutively sampled, starting from one measurement for each of 24 plots (24 samples) to all validation samples. This series of sampling steps allow checking the method performance in relation to the number of reference measurements. This allowed us to conclude that the number of reference measurements is crucial for the described moisture estimation improving method. Those measurements were carried out using the gravimetric method in the study, and all 24 sensors on 24 different plots were calibrated simultaneously. In addition, the necessary number of readouts for such a setup has been investigated.

The following Figure 5 presents the compilation of the performance for models trained on the mentioned subsampled training sets. The analysis of the *x*-axis allows one to address the question: how many reference measurements are necessary to obtain satisfactory moisture estimation performance. Keeping in mind that this number corresponds to the calibration of 24 sensors, one could spot that 300 reference measurements have reduced the median calibration error already below 1%.

## 4. Discussion

The comparison of moisture measurements obtained in various ways was made by many authors, among others, to obtain reliable data with a lower workload or money.

Ventura and co-authors [13] compared ECH2O probes, EC-5, and time domain reflectometer sensors (CS616), with gravimetric data. According to their results, sensors can be used in each type of soil with the same calibration equation, independently from depth, with RMSE range between 2.5 and 3.6%.

The study objective of another group of authors [27] was to develop a calibration methodology for the YL-69 sensor and evaluate its accuracy with two commercial ECH2O probes. After such a process, the YL-69 sensor performed well for tested soil types and let the researchers obtain a coefficient of determination value ($R^{2}>0.90$) and root mean square error (RMSE ≤0.01). After precise method application, the YL-69 sensor performed as well as the EC-5 ($R^{2}=0.92$) and 5TE probes ($R^{2}=0.98$). The relatively low cost of such YL-69 sensors sets an attractive alternative for the popular EC-5 and 5TE sensors in determining the water content in the soil.

The investigation of Serrano and co-authors [28] aimed at finding a decent indirect method for estimating surface soil moisture that could be applied for monitoring in situ soil water potential. They measured the surface volumetric soil moisture by the gravimetric method and collected 2211 soil samples for the survey period. Their results show a nonlinear relationship between gravimetric soil measurement and the indirect estimation displaying a determination coefficient of 68.4%.

The following researchers, Olszewska and Nowicka [7], compared the performance of a similar method on two different soil profiles, utilizing the volumetric moisture measurement with a TDR method and gravimetric method and finishing with a linear correlation $R^{2}=0.71$.

Verma and Pahuja [8] investigated the laboratory-based virtual instrumentation (VI) and used the evaluated system to recalibrate various smart soil moisture sensors, and provided modeling and performance analysis for these sensors as well. Their goodness of fit of the model’s analysis revealed that polynomial models of both the sensors are more accurate ($R^{2}$ = 0.96:0.98) than simple linear models ($R^{2}=$ 0.90:0.96).

For the final model in the presented study, the coefficient of determination $R^{2}$ was 0.858. This result is slightly lower than some of the methods discussed above. Still, it should be remembered that it was verified based on actual measurements and determined simultaneously for 24 different sensors and measurements carried out for many weeks under changing conditions, which is a particular difficulty for a direct comparison of these results. For a reliable assessment of the discussed results of the humidity estimation, it is worth mentioning the estimation error, which was only 0.808%.

The presented method has been found performant using the studied validation set of measurements, but due to its requirements, it forces the potential user to provide the strict data. The collection of referencing measurements and the additional meteorological readouts could sometimes limit its usage. Additionally, the future method user should consider the number of referencing measurements, as it substantially influences the final performance. As we tried to point out in Section 3.3, this is related to the number of sensors one would like to support using such estimation methodology.

## 5. Conclusions

The above-presented method of improving the accuracy of soil moisture measurement at the root level seems to be an advantageous way to obtain precise measurements at a low cost. It reduces measurement errors from a few to less than one percent without replacing inexpensive sensors with more advanced models. The use of sensors for soil moisture measurements is a convenient method. The improvement of their readings with the presented models allows for obtaining a satisfactory accuracy of the estimated measurements. The presented method allowed for reducing the evaluated median absolute error from 2.035 to 0.831 (59% error shrinkage) compared to the baseline model. It should ultimately allow for building a large set of measurements and can be an attractive alternative to the time-consuming and costly individual calibration of each sensor or the purchase of more advanced sensors.

The comparison of the tested models’ results let us test several features that were found helpful in terms of model performance improvement and interpretability. Those features, besides the sensor measurements (the most significant 57.4% importance for the final model), were the watering strategy and the soil indicator, meteorological parameters (air humidity and temperature) and the current soil temperature measured at different depths, and to some extent, the amount of watering (accumulated for the past hour) and measured rainfall (for the past six hours). The main determinator of the resulting estimator was the number of the reference gravimetric measurements that significantly influenced the final model performance.

On the other hand, the necessity to collect the reference measurements is a certain limitation in applying this method. The indispensable dataset has to be prepared, e.g., using the traditional gravimetric method, and requires the monitoring of meteorological parameters near the measurement site—because, due to these measurements presented, ensemble learning models obtain the ability to provide precise estimates.

The measurements used in the study were collected to determine the optimal amount of water doses in the experimental plots irrigation. Therefore, some of the measurements were made in situations where the conditions were relatively dry—watering is not carried out when there is heavy rainfall. However, such a special nature of the measurements did not allow for testing the behavior of the model in situations of high soil moisture, which would probably be worth verifying in the next stages.

The work on the presented method could also be extended by verifying models in the longer term and in subsequent seasons, which should allow for checking the method’s operation and for conditions other than those in which the measurements were used. In subsequent experiments, it would be also worth examining other features or data sources that could possibly help improve the estimation of humidity.

The next planned directions of the study are aimed at using some form of non-contact, remote measurements, e.g., with the use of imaging, which could increase the spatial resolution of the measurements carried out. For the estimation carried out with the use of data collected by unmanned aerial vehicles or based on satellite data, the result of such estimation could ultimately generate humidity maps, which would be a high-quality supplement to the method.

## Figures and Tables

**Figure 1 sensors-22-04591-f001:**
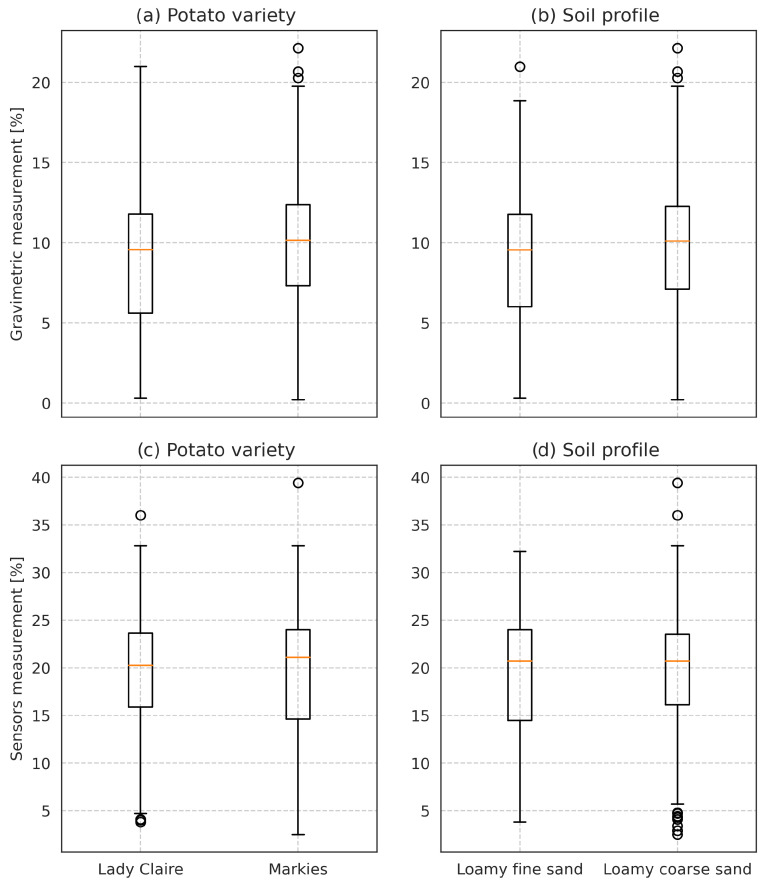
Moisture measurements’ subsets distribution, according to: variety for Lady Claire and Markies, according to: soil profile for loamy fine sand and loamy coarse sand separately, (**a**,**b**) presents gravimetric measurements; (**c**,**d**) presents sensor measurements. The box on the plots denotes quartiles (Q1, Q3), the middle horizontal line refers to the median, and the whiskers are calculated using the interquartile range, but drawn to the largest measurements within the calculated whisker. Measurements outside the whiskers are marked as outliers using circles.

**Figure 2 sensors-22-04591-f002:**
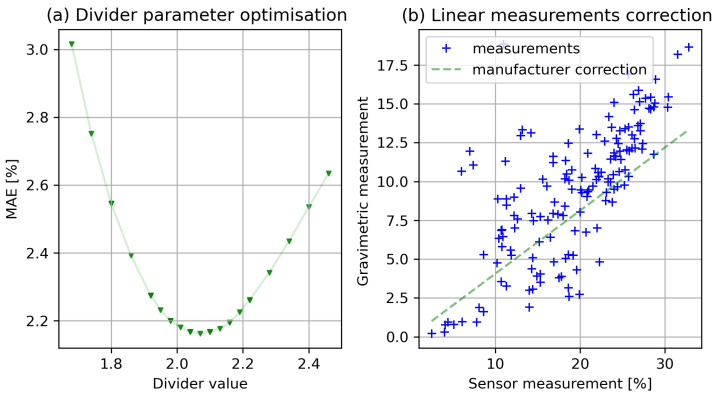
Initial linear model for root moisture sensor calibration; (**a**) the improvement in the model performance after the linear divider optimization; (**b**) the visualization of the resulting linear model and the reference gravimetric measurements.

**Figure 3 sensors-22-04591-f003:**
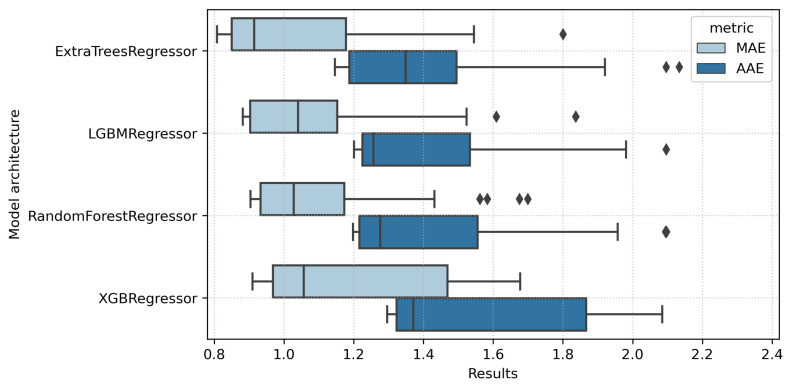
Different model architecture performance comparisons for all tested feature configurations.

**Figure 4 sensors-22-04591-f004:**
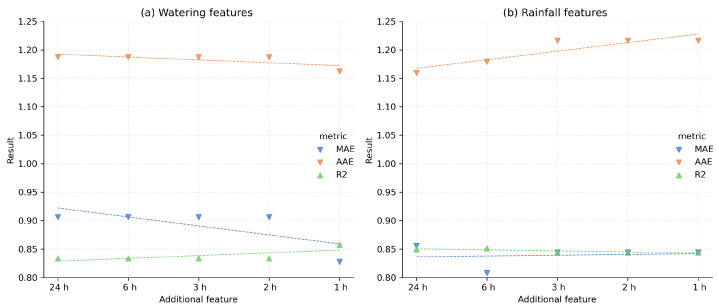
The comparison of the different models with additional information on watering and measured rainfall and with different time-frames when those features were analyzed. Compared results (*y*-axis) were calculated for different models (*x*-axis) with information on watering or measured rainfall during the past: 1, 2, 3, 6, or 24 h.

**Figure 5 sensors-22-04591-f005:**
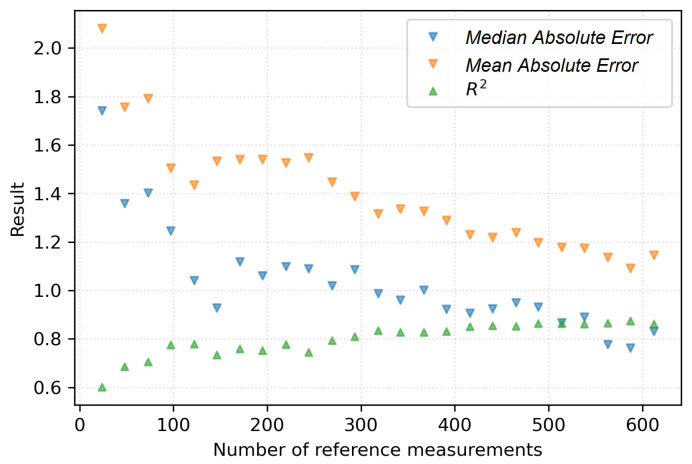
Model performance results for models trained on a narrowed reference measurements set.

**Table 1 sensors-22-04591-t001:** Watering amounts during the experiment.

Plot Number	Water Amount (mm)	Number of Water Treatments
1, 2, 13, 14	6	1
7, 8, 19, 20	4	1
3	16	2
4	33	4
5	44	4
6	53	4
9	19	2
10	53	7
11	73	7
12	89	7
15	10	2
16	34	4
17	51	4
18	47	4
21	12	2
22	46	7
23	82	7
24	80	7

**Table 2 sensors-22-04591-t002:** Results for models with additional training features (to distinguish different soil profiles or each experimental plot using its number as a feature). The best results are in bold font.

Model	MAE	AAE	RMSE	R${}^{2}$	Features
Baseline Linear Model	2.035	2.545	3.291	0.384	moisture probe
Ridge	1.641	2.188	2.950	0.505	moisture probe
Extra Trees	1.545	2.134	2.818	0.548	moisture probe, soil type
Extra Trees	**1.462**	**1.891**	**2.490**	**0.647**	moisture probe, field

**Table 3 sensors-22-04591-t003:** Performance of the different models trained using different training sets comprised of several features’ configurations. For each set, the best performing model architecture, characterized by the lowest median absolute error, is listed in the first column. Metrics that are listed in brackets were determined using the training set. The best results are in bold font.

Model	MAE	AAE	RMSE	R${}^{2}$	Features
XGB	1.471 (0.334)	1.937 (0.543)	2.575 (0.890)	0.623 (0.951)	moisture probe, field, soil type
LGBM	1.044 (0.829)	1.488 (1.189)	2.060 (1.799)	0.758 (0.801)	moisture probe, H200, T200, Tg5, Tg10, Tg20, Tg50
LGBM	1.060 (0.828)	1.477 (1.221)	2.050 (1.827)	0.761 (0.795)	moisture probe, H200, T200, Tg5
XGB	0.999 (0.994)	1.466 (1.360)	2.051 (1.909)	0.761 (0.776)	moisture probe, H200, T200, Tg10
Random Forest	1.078 (0.978)	1.545 (1.444)	2.066 (2.118)	0.757 (0.725)	moisture probe, H200, T200, Tg20
Extra Trees	1.111 (0.920)	1.405 (1.373)	1.992 (2.021)	0.774 (0.749)	moisture probe, H200, T200, Tg50
Extra Trees	0.914 (0.292)	1.349 (0.413)	1.876 (0.641)	0.800 (0.927)	moisture probe, field, watering type, soil type, houses, Tg5
Extra Trees	**0.831** (0.278)	**1.146** (0.383)	**1.558** (0.573)	**0.862** (0.961)	moisture probe, field, watering type, soil type, houses, H200, T200, Tg5, Tg10, Tg20, Tg50

**Table 4 sensors-22-04591-t004:** Results for Extra Trees models with additional features on watering or rainfall. Results presented in consecutive rows correspond to different watering or rainfall cumulating time-frames. The best results are in bold font.

Additional Feature	AAE	MAE	$R^{2}$	RMSE
watering 1 h	**1.162**	0.828	**0.858**	**1.580**
watering 2 h	1.187	0.906	0.834	1.709
watering 3 h	1.187	0.906	0.834	1.709
watering 6 h	1.187	0.906	0.834	1.709
watering 24 h	1.187	0.906	0.834	1.709
rainfall 1 h	1.216	0.844	0.844	1.655
rainfall 2 h	1.216	0.844	0.844	1.655
rainfall 3 h	1.216	0.844	0.844	1.655
rainfall 6 h	1.179	**0.808**	0.852	1.615
rainfall 24 h	1.159	0.856	0.849	1.626

## Data Availability

Not applicable.

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
