# Peer review of "Soil Moisture a Posteriori Measurements Enhancement Using Ensemble Learning"

_sensors, 2022, doi:10.3390/s22124591_

Round 1
Reviewer 1 Report
This manuscript presented an interesting study on characterizing the commonly used smart soil-moisture sensors by integrating ensemble learning techniques. In general, the manuscript is well-written and moderate revision is suggested. Some technical issues can be addressed or clarified to improve the quality of the manuscript:
(1) The introduction containing 8 paragraphs tends to be relatively fragmented, which poses difficulties in capturing the necessity of this research. It is advisable to reorganize this section and highlight the contribution of this study compared with previous studies.
(2) Subsection 3.2 compared the predictive performance of the four machine learning algorithms, namely XGB, LGBM, RF, and extra trees. Whether the results presented in Table 3 calculated from the training dataset or testing dataset. Both the results on the training dataset and testing dataset are suggested to be provided.
(3) The detailed descriptions of the four machine learning algorithms were omitted in the manuscript probably for brevity. To improve the readability of these algorithms (e.g., RF and XGB), it is advisable to provide some related references, and thus interesting readers can refer to the cited references, such as ‘Efficient reliability analysis of earth dam slope stability using extreme gradient boosting method’ and ‘Slope stability prediction using ensemble learning techniques: A case study in Yunyang County, Chongqing, China’.
(4) The machine learning algorithms generally contain several hyperparameters, and several hyperparameter optimization techniques have been developed, such as the Bayesian optimization technique in ‘Prediction of undrained shear strength using extreme gradient boosting and random forest based on Bayesian optimization’. Please specify which optimization technique was used in this study.
(5) Table 3 compared the predictive performance of different modesl with different features. It can be observed that extra tree model performed the best among them, please explain the underlying reasons.
(6) A total of 11 features were used in the construction of extra tree model (i.e., the last row of the last column in Table 3). In machine learning, feature important analysis is frequently applied to explore the relative contribution of each input to the output, such as. It is suggested to add the feature importance analysis result, further helping engineers focus on the more influential parameters in practical applications.
(7) The section ‘5. Conclusions’ is relatively lengthy, it is suggested to focus more on the implications of this study for geotechnical researchers and practitioners in the soil moisture measurement.
Author Response
Dear Reviewer,
We accept all points raised in the revision.
We rewrote several fragments and updated the description according to the reviewers' suggestions. We appended some new sentences and citations.
Please find enclosed the revised manuscript, which includes editorial comments, more than a dozen new paragraphs, and several additional references.
Yours faithfully,
Bogdan Ruszczak and Dominika Boguszewska-Mańkowska
(1) The introduction containing 8 paragraphs tends to be relatively fragmented, which poses difficulties in capturing the necessity of this research. It is advisable to reorganize this section and highlight the contribution of this study compared with previous studies.
We followed the advice, to improve the manuscript we: rewrote several sentences, rearranged 4 paragraphs, removed one paragraph, and appended several references.
(2) Subsection 3.2 compared the predictive performance of the four machine learning algorithms, namely XGB, LGBM, RF, and extra trees. Whether the results presented in Table 3 calculated from the training dataset or testing dataset. Both the results on the training dataset and testing dataset are suggested to be provided.
The suggested performance metrics gained using the training set have been provided.
(3) The detailed descriptions of the four machine learning algorithms were omitted in the manuscript probably for brevity. To improve the readability of these algorithms (e.g., RF and XGB), it is advisable to provide some related references, and thus interesting readers can refer to the cited references, such as ‘Efficient reliability analysis of earth dam slope stability using extreme gradient boosting method’ and ‘Slope stability prediction using ensemble learning techniques: A case study in Yunyang County, Chongqing, China’.
Thank you for the interesting lecture suggestion that we found interesting, paper related, and included in our algorithm description, as well as some additional citations. Thus, the manuscript has been supplemented with several new references.
(4) The machine learning algorithms generally contain several hyperparameters, and several hyperparameter optimization techniques have been developed, such as the Bayesian optimization technique in ‘Prediction of undrained shear strength using extreme gradient boosting and random forest based on Bayesian optimization’. Please specify which optimization technique was used in this study.
We appended the citation to the mentioned method, and also a short comment to the manuscript on the training procedure we used.
The Following fragments have been appended:
[...] It is worth recommending a proper training procedure implementing cross-validation, or a prominent optimization method, to determine the appropriate model hyper-parameters, i.e. following the procedure \cite{WengangChongzhi21} that is commonly reported as performance improvements. To prepare models for this investigation, the standard five-fold cross-validation procedure using the training set has been used.
(5) Table 3 compared the predictive performance of different modesl with different features. It can be observed that extra tree model performed the best among them, please explain the underlying reasons.
Thank you for this suggestion. The Following fragments have been appended:
[...] For the models based on Extra Trees results were found better for the investigated validation set. This could mean the algorithm had a lower tendency to overfit to training data, and in this case has better generalization ability.
(6) A total of 11 features were used in the construction of extra tree model (i.e., the last row of the last column in Table 3). In machine learning, feature important analysis is frequently applied to explore the relative contribution of each input to the output, such as. It is suggested to add the feature importance analysis result, further helping engineers focus on the more influential parameters in practical applications.
We followed this suggestion and appended the following fragment, regarding feature importance to the manuscript:
A piece of interesting information occurs when one looks closely at the features' importance values. As expected, the most important is the variable denoting probe readout (moisture_probe) which value is 57.4%. Similarly important are the watering type (8.6%) and soil temperature at 50 cm depth (Tg50 – 7.7%). Several others ranging from 4 – 5% (Tg20=4.9%, Tg10=4.6%, Tg5=4.2%), and some others with the lower value. But surprisingly low importance was related to the soil type mapping feature (soil_type), with the lowest value equal to only 0.8%.
(7) The section ‘5. Conclusions’ is relatively lengthy, it is suggested to focus more on the implications of this study for geotechnical researchers and practitioners in the soil moisture measurement.
We followed the advice, rewritten several sentences in that part, and added comments on the results and their interpretations as well as the potential meaning of modeled parameters. Our difficulty is there are several profiles of readers we aim for and we hope to address many of them with paper limitations.
Reviewer 2 Report
The manuscript presents interesting study on improvement of soil moisture measurements based on ensemble learning. The manuscript is quite well prepared, however contains some drawbacks.
Below are detailed comments:
1) Introduction contains information which are not supported by references. There are only 4 references cited. For example if there is information about 10HS sensor, appropriate reference should be cited.
2) Names of soil, i.e. heavy clay sand, medium clay, light loamy sand, and light clay are not according international standard. I suggest to use soil texture classes according international standards, such as eg. USDA: https://www.nrcs.usda.gov/wps/portal/nrcs/detail/soils/survey/?cid=nrcs142p2_054167
3) Line 112: Please add approximate geographical coordinates of the location.
4) Fig. 1. Please explain what is presented at the box-plots. Is it median, quartiles and range? It should be clearly stated. How the outliers in these figures were selected?
5) Fig. 2. In the Fig. 2b linear measurement correction is based on linear regression with intercept equal 0. Visual evaluation of the Fig. 2b allows the conclusion that a better regression function would be the function with intercept below 0 because it would be better fitted function. I suggest to fit such a regression function.
6) It is not clear what is as a “additional feature” and “result” in the Fig. 4 in the description of the horizontal and vertical axes. Please notice that figures and tables should be self-explanatory, ie. clear enough without reading the text of the manuscript.
7) Line 327: Please use dot not comma as a decimal separator.
Author Response
Dear Reviewer,
We accept all points raised in the revision.
We rewrote several fragments, updated the description according to the reviewers' suggestions. We appended some new sentences and citations.
Please find enclosed the revised manuscript, which includes editorial comments, more than a dozen new paragraphs, and several additional references.
Yours faithfully,
Bogdan Ruszczak and Dominika Boguszewska-Mańkowska
1) Introduction contains information which are not supported by references. There are only 4 references cited. For example if there is information about 10HS sensor, appropriate reference should be cited.
Thank you for this suggestion. To make it more clear we supplemented several additional references to this part.
2) Names of soil, i.e. heavy clay sand, medium clay, light loamy sand, and light clay are not according international standard. I suggest to use soil texture classes according international standards, such as eg. USDA: https://www.nrcs.usda.gov/wps/portal/nrcs/detail/soils/survey/?cid=nrcs142p2_054167
We followed this suggestion and changed the naming of the soil profiles we used in the study according to the mentioned standard. In all 3 places (lines: 90-91, 124, Figure 1, Figure 1 caption) in the paper, we follow those names: loamy fine sand, loamy coarse sand.
3) Line 112: Please add approximate geographical coordinates of the location.
Thank you for this suggestion. We added the sentence to the mentioned fragment:
The geographical coordinates of the plots that were used in the study are: N 52° 29.00562 E 21° 2.67360
4) Fig. 1. Please explain what is presented at the box-plots. Is it median, quartiles and range? It should be clearly stated. How the outliers in these figures were selected?
Thank you for this suggestion. To make it more clear we extended the mentioned figure label with the following explanations:
The box on the plots denotes quartiles (Q1, Q3), the middle horizontal line refers to the median, and the whiskers are calculated using the interquartile range, but drawn to the largest measurements within the calculated whisker. Measurements outside the whiskers are marked as outliers using circles.
5) Fig. 2. In the Fig. 2b linear measurement correction is based on linear regression with intercept equal 0. Visual evaluation of the Fig. 2b allows the conclusion that a better regression function would be the function with intercept below 0 because it would be better fitted function. I suggest to fit such a regression function.
We agree that the intercept parameter should also be fitted, with better performance. For the baseline model, we followed the advice of the sensor manufacturer and used this very simple linear model. We did not perform this, to check what is the performance of this baseline model, but just at the beginning of the commented study. Models presented in the other part of the paper do not have that limitation, and as you anticipated, provide better estimation performance. The models that were presented in the following stages of the experiment do not fix to intercept with point (0,0).
6) It is not clear what is as a “additional feature” and “result” in the Fig. 4 in the description of the horizontal and vertical axes. Please notice that figures and tables should be self-explanatory, ie. clear enough without reading the text of the manuscript.
Thank you for this suggestion. To make it more clear we extended the mentioned figure label with the sentence:
Compared results (Y-axis) were calculated for different models (X-axis) with information on watering or measured rainfall during the past: 1, 2, 3, 6, or 24 hours.
7) Line 327: Please use dot not comma as a decimal separator.
The suggested separator has been fixed accordingly.
Reviewer 3 Report
The article is well presented in scientific terms and the reviewer believes that it will be of interest to its readers.
The reviewer proposes that the article be accepted for publication.
The reviewer has some remarks and recommendations:
1. Table 1 - the abbreviation "ID" should be written in words.
2. Comments on the results of Table 1 may be provided.
3. 186 line: to give the formula for determining the error AAE.
4. 191- 195 line: to give the abbreviations of the described parameters and the formulas by which they are calculated.
5. 212 line: MEA - the abbreviation is already given on 187 line and does not need to be repeated.
6. 212 line: AAE - it is better to give this abbreviation to line 186, where it occurs for the first time.
7. lines 307-332: this part of the article is better to be presented in the literature review. In the sections results and discussions it is good to emphasize the results obtained by the authors and to make a comparative (preferably tabular) analysis with better research.
8. According to the reviewer, the chosen model should be described in detail, and a scheme can be applied.
9. What are the limitations of the modeling and the conducted research?
10. The conclusions made in the last section should be supported by the obtained numerical results.
Author Response
Dear Reviewer,
We accept all points raised in the revision.
We rewrote several fragments and updated the description according to the reviewers' suggestions. We appended some new sentences and citations.
Please find enclosed the revised manuscript, which includes editorial comments, more than a dozen new paragraphs, and several additional references.
Yours faithfully,
Bogdan Ruszczak and Dominika Boguszewska-Mańkowska
- Table 1 - the abbreviation "ID" should be written in words.
The suggested column name of Table 1 has been renamed to be more informative.
- Comments on the results of Table 1 may be provided.
The suggested section was supplemented with additional comments on the mentioned table.
To provide various soil moisture readouts during the experiment, different watering has been applied. In Table, one can find the exact amount that was delivered to each microplot. It is crucial to prepare a set of measurements of high variance that will later allow proper method evaluation and preparation of a performant algorithm.
- 186 line: to give the formula for determining the error AAE.
The metric mentioned in 186 line (AAE) is just standard, popular mean absolute value, so we thought for a while why you asked for this specific one to be written, and we found our obvious, redaction mistake. I was wrongly written as average (not absolute). We corrected this mistake in the manuscript.
- 191- 195 line: to give the abbreviations of the described parameters and the formulas by which they are calculated.
The suggested abbreviations have been supplemented in the manuscript in the mentioned lines (MSE, RMSE, ME, MSLE).
- 212 line: MEA - the abbreviation is already given on 187 line and does not need to be repeated.
According to the suggestion, the repeated information has been removed.
Line: “To present the results, we used the following abbreviations: MAE and 211
AAE [....]” dropped off.
- 212 line: AAE - it is better to give this abbreviation to line 186, where it occurs for the first time.
We followed this suggestion and updated the manuscript accordingly.
- lines 307-332: this part of the article is better to be presented in the literature review. In the sections results and discussions it is good to emphasize the results obtained by the authors and to make a comparative (preferably tabular) analysis with better research.
The mentioned section has been rewritten to improve readability.
- According to the reviewer, the chosen model should be described in detail, and a scheme can be applied.
We followed this suggestion and appended several fragments to the manuscript, in the part describing the proposed model, with comments on how the final model is build, and how to get the estimations using the model.
The procedure required to achieve the eventual estimations for probe readouts, besides those readouts themselves (a model feature called: moisture probe) requires providing more data. Feeding the calculation with air temperature, air humidity, and soil temperature (model features: (H200, T200, Tg5, Tg10, Tg20, Tg50) is necessary. It also needs values related to the exact experimental plot, like field number, that identifies the probe, type of soil, and watering treatment type (model features: field, soil type, watering type, houses).
A piece of interesting information occurs when one looks closely at the features' importance values. As expected, the most important is the variable denoting probe readout (moisture_probe) which value is 57.4%. Similarly important are the watering type (8.6%) and soil temperature at 50 cm depth (Tg50 – 7.7%). Several others range from 4 – 5% (Tg20=4.9%, Tg10=4.6%, Tg5=4.2%), and some others with the lower value. But surprisingly low importance was related to the soil type mapping feature (soil_type), with the lowest value equal to only 0.8%.
- What are the limitations of the modeling and the conducted research?
Thank you for the suggestion, we added several sentences, addressing those limitations to this section.
- The presented method has been found performant using the studied validation set of measurements, but due to its requirements, it forces the potential user to provide the strict data. The collection of referencing measurements and the additional meteorological readouts could sometimes limit its usage.
- Additionally, the future method user should consider the number of referencing measurements, as it substantially influences the final performance. As we tried to point out in section 3.3. this is related to the number of sensors one would like to support using such estimation methodology.
- The conclusions made in the last section should be supported by the obtained numerical results.
Thank you for this suggestion, we’ve included additional comments with results to the mentioned section. I.e.:
[...] the presented method allowed to reduce the evaluated median absolute error from 2.035 to 0.831 (59% error shrinkage) compared to the baseline model.
[...]Those features, besides the sensor measurements (the most significant 57.4% importance for the final model),...
Round 2
Reviewer 1 Report
Thanks to the contribution of all the authors, the manuscript has been carefully revised according to the reviewer comments. In general, the accept is suggested.
Author Response
Thank you for the revision, we have accepted all the suggestions and applied changes accordingly.
Best wishes
Authors
Reviewer 2 Report
All my comments were included in the corrected manucript.
I have one doubt connected with soil profile. Soil texture is changing depending on soil layer, e.g. for layer 0-30 cm in the same place, soil texture can be different. Additional information about soil type (e.g according FAO WRB) should be added for soil description.
Author Response
Thank you for the suggestion.
For this study, we investigated only topsoil, and this is described in the paper accordingly to the classification you've previously mentioned. The sensors for the described measurement setup were located in the top layer. Thus we did not analyze lower soil layers.
Best wishes
Authors